# Childhood Anemia in Mozambique: A Multilevel Mixed-Effects Analysis of 2011–2022/23 Population-Based Surveys

**DOI:** 10.3390/healthcare13060635

**Published:** 2025-03-14

**Authors:** Réka Maulide Cane, Rornald Muhumuza Kananura, Ronald Wasswa, Maria Patrícia Gonçalves, Luís Varandas, Isabel Craveiro

**Affiliations:** 1Instituto Nacional de Saúde (INS), Ministério da Saúde (MISAU), Estrada Nacional EN1, Bairro da Vila—Parcela No 3943, Distrito de Marracuene, Marracuene 264, Província de Maputo, Mozambique; mariapatriciagoncalves@gmail.com; 2Unidade de Ensino e Investigação de Saúde Pública Global, Global Health and Tropical Medicine (GHTM), Associate Laboratory in Translation and Innovation Towards Global Health (LA-REAL), Instituto de Higiene e Medicina Tropical (IHMT), Universidade Nova de Lisboa (UNL), Rua da Junqueira 100, 1349-008 Lisboa, Portugal; 3African Population and Health Research Center (APHRC), Nairobi P.O. Box 10787-00100, Kenya; mk.rornald@yahoo.com; 4Centre of Excellence for Maternal and Newborn Health, Makerere University School of Public Health, Kampala P.O. Box 7072, Uganda; rwasswa93@yahoo.com; 5Department of Health Policy Planning and Management, Makerere University School of Public Health, New Mulago Complex, Kampala P.O. Box 7072, Uganda; 6Unidade de Ensino e Investigação de Clínica das Doenças Tropicais, Global Health and Tropical Medicine (GHTM), Instituto de Higiene e Medicina Tropical (IHMT), Universidade Nova de Lisboa (UNL), Rua da Junqueira 100, 1349-008 Lisboa, Portugal; varandas@ihmt.unl.pt; 7Hospital Dona Estefânia, Centro Hospitalar Universitário Lisboa Central, 1150-199 Lisboa, Portugal; 8NOVA Medical School, Faculdade de Ciências Médicas Universidade Nova de Lisboa, Campo Mártires da Pátria 130, 1169-056 Lisboa, Portugal

**Keywords:** anemia, childhood, determinants, population-based surveys, Mozambique, sub-Saharan Africa

## Abstract

**Background/Objectives**: Anemia adversely affects children’s cognitive and motor development and remains a global public health problem. This study aimed to identify the individual, feeding, household, and community determinants of anemia among children in Mozambique. **Methods:** We used pooled datasets of two Mozambique representative population-based surveys: the 2011 and 2022–2023 Demographic and Health Surveys. A total sample of 8143 children aged 6–59 months with available hemoglobin testing was included. Multilevel mixed-effects analysis was performed using STATA (18.0). **Results:** Over a decade, the prevalence of anemia in children aged 6–59 months remained high, increasing slightly from 69.1% in 2011 to 72.9% in 2022. Children aged 6–11 months were less likely to have anemia than children from other age groups (aOR = 0.77, 95% CI = 0.62–0.96). Children who suffered from illnesses (aOR = 1.44, 95% CI = 1.18–1.75), received vitamin A supplements (aOR = 0.76, 95% CI = 0.63–0.93), lived in female-headed households (aOR = 1.16, 95% CI = 1.01–1.32), and who lived in households with unimproved drinking water sources (aOR = 1.40, 95% CI = 1.19–1.65) were more likely to have anemia than their peers. Overall, 16% of the variability in anemia prevalence was attributed to differences between clusters (ICC = 0.16). **Conclusions:** Childhood anemia remains a critical public health challenge in Mozambique, with prevalence rates exceeding the average for sub-Saharan Africa. Multisectoral approaches to enhance essential supplies’ provision and the primary healthcare monitoring of children at risk favored more investments in rural development and sustainable agriculture, water sanitation, and social care and gender-sensitive work policies that can help tackle childhood anemia.

## 1. Introduction

Anemia is the most common hematologic abnormality identified in infants and children and it adversely affects cognitive and motor development and causes fatigue and low productivity, influencing country development [1,2,3,4]. Anemia is associated with increased morbidity and mortality in children, particularly children of preschool age [2]. Anemic children from disadvantaged communities often fail to achieve their age-related potential, having learning problems and inattentiveness [5]. While iron deficiency is one of the most important factors of anemia, causes for the condition are often multifactorial, including other micronutrient deficiencies (such as vitamin B12 and folic acid), infections (e.g., malaria and HIV), and inherited blood diseases. Social and cultural factors may also predispose to anemia [1,6,7,8]. The most common causes of iron deficiency in children include chronic blood loss and poor diets (e.g., nutrient-poor diets, severe restriction diets, and excessive daily cow’s milk consumption that decreases iron absorption) [9,10]. Several micronutrients (such as vitamins A, B12, and folic acid) are required for the normal production of red blood cells [11]. Thus, micronutrient deficiencies often caused by poor dietary quality (inadequate micronutrient intake) and infectious diseases can lead to childhood anemia by affecting iron metabolism [11,12,13]. Infectious diseases may cause anemia through several mechanisms, such as poor appetite and food intake, decreased red blood cell production, hemolysis, and blood and iron loss [11]. Children with inherited blood diseases are more prone to infections and malnutrition and hence more susceptible to hemolysis, which can lead to anemia [11]. Environmental, social, and cultural factors may also predispose to anemia [1,6,7,8,9,11]. Low socioeconomic status and poor access to healthcare services are often linked to poor and unsanitary living conditions, poor nutrition, and limited access to nutritional supplements, contributing to an increased risk of anemia [11]. In addition, cultural and economic factors influence food selection and household food distribution and can pose a barrier to dietary modification, particularly in low–middle-income countries [14].

Globally, about 39.8% of children aged 6–59 months are affected by this condition. Low and middle-income countries bear the highest burden, and anemia risk has not considerably improved over the past two decades across most of Africa. In 2019, the prevalence of anemia in children under five was highest in the African Region (with a prevalence rate of 60.2%) [3,7]. Sub-Saharan Africa endures a substantial burden of anemia (overall prevalence rate of 64.1%), showing prevalence variation across the east (56.0%), west (62.2%), central (75.1%), and southern (50.7%) regions [9]. Thus, the targets of the Sustainable Development Goals (SDGs) for anemia reduction might be beyond reach by 2030. The WHO Africa region remains among the most affected, with about 103 million children affected by anemia [3,7]. Anemia serves as a country’s growth indicator as it is positively associated with measures of human capital index indicators (such as undernutrition and under-five mortality), with long-term consequences including decreased physical growth and lower educational attainment, cognition, workforce productivity, and wages. Thus, for countries to improve the future economic development of their economies, it is crucial to address the drivers of anemia [15,16,17,18,19,20]. Despite ongoing efforts to address nutritional anemia among children—such as mass deworming, iron supplementation, emergency food assistance, food vouchers, and nutrition counseling—food insecurity persists [21]. As a result, Mozambique continues to be one of the sub-Saharan African countries with a high burden of childhood anemia, ranking 13th in league tables [21]. Few studies have been conducted on childhood anemia in Mozambique using data from population-based surveys [21,22]. Nonetheless, these studies did not use a pooled dataset from various national surveys, and did not perform a comprehensive pooled data analysis including key determinants such as children’s feeding characteristics. By addressing these gaps, this study provides new insights of critical importance for local policymakers and key stakeholders in anemia prevention and management in Mozambique.

This study used a pooled dataset from two national surveys to identify the individual, feeding, household, and community determinants of childhood anemia in Mozambique.

## 2. Materials and Methods

### 2.1. Study Area

The study area of this research was the Republic of Mozambique, located in Southeast Africa. With a long Indian Ocean coastline of 2700 km, Mozambique is bordered by South Africa, Eswatini, Tanzania, Malawi, Zambia, and Zimbabwe, facing east Madagascar [23]. In 2022, Mozambique’s population was 33 million people, and it is projected to grow more than double by 2050, to 67.8 million people—which shows the need to address childhood anemia now rather than later [24,25]. The Mozambican population is very young and about two-thirds live and work in rural areas [23,25]. Mozambique has eleven provincial capital cities, including its capital—Maputo City. Mozambique’s Human Development Index is considerably low (185th out of 191 countries in 2021), and the level of education among the Mozambican population is also low, with an average of 4.5 years of schooling for men and 2.7 years for women. The country is often susceptible to natural disasters (cyclones, floods, and droughts) which disrupt the country’s economic development [24,25,26,27]. During natural disasters, poor sanitary conditions and limited access to basic commodities often lead to outbreaks of diseases, particularly, malaria, diarrhea, cholera, and dysentery, and malnutrition resulting from food insecurity is also common [27].

### 2.2. Data Source

This study used data from the 2011 Demographic and Health Survey (DHS), and the 2022–2023 Demographic and Health Survey (DHS). Data were obtained after administrative approval through the DHS Program website (https://dhsprogram.com/, accessed on 21 September 2023). The DHS 2022–2023 and DHS 2011 provide information on the health and socio-economic characteristics of the interviewed population, including aspects related to anemia in children under 5 years. More details regarding the methodology used in DHS 2022–2023 and DHS 2011 can be found in survey reports publicly available [28,29].

The 2011 DHS comprised a probabilistic, stratified, and multi-stage sample, selected from the 3rd General Population and Housing Census of Mozambique (RPGH 2007), conducted by the Instituto Nacional de Estatística (INE) in 2007. For this survey, in the first stage, 611 PSUs (Primary Sampling Units) were selected, with probability proportional to size, with size being the measure of the number of households in each stratum within each province. In the second sampling stage, 20 households in urban PSUs and 25 households in rural PSUs with equal probabilities were selected. In the third stage, exhaustive sampling was carried out, and demographic and health data were collected from all women aged 15–49 and children under 5 years old found in the selected households [30].

As for the 2022–2023 DHS, the sample design consisted of two stages. First, a sample of clusters was selected, consisting of EAs defined for the population based on the 4th General Population and Housing Census (RGPH 2017), conducted by the INE in 2017. A total of 619 EAs were selected, with probability proportional to size, with size being the measure of the number of households in each explicit stratum. Of the 619 EAs, 232 were from urban areas and 387 from rural areas. Due to security concerns, eight districts (Ibo, Macomia, Mocímboa da Praia, Mueda, Muidumbe, Nangade, Palm, and Quissanga) in Cabo Delgado province were excluded from the sample selection. In the second stage, 26 households were systematically selected, with equal probabilities from each enumeration area. Based on this procedure, 16,045 households were selected. This number is slightly smaller than the sample size of 16,094 because two selected EAs (one in Cabo Delgado and one in Zambézia Province, both rural) could not be completed due to security issues [31]. The DHS 2022–2023 and DHS 2011 surveys included anemia testing in children aged 6–59 months. For all surveys, testing was performed using the HemoCue^®^ 201+ for measuring the amount of hemoglobin in the blood. Informed consent from the children’s guardians was requested before collecting blood samples for anemia testing [28,29]. This study analyzed a weighted subpopulation of children aged 6–59 months born (and alive) at the time of the survey with available data on hemoglobin testing (n = 8143).

### 2.3. Selection of Study Participants

In the DHS 2011 and 2022–2023, child anemia measurements were performed only in children aged 6–59 months [28,29]. As such, this study included information on children (a) who were aged 6–59 months (at the time of the surveys) and (b) who had undergone anemia testing and had data available on hemoglobin (Hb) determination.

### 2.4. Variables of the Study

In the Appendix A displays additional information on study outcome and exposure variables. In the Appendix A shows a conceptual framework for child anemia determinants and consequences. The selection of exposure variables was guided by evidence from the literature which indicates that such determinants are important factors to be considered for anemia [21,32,33,34,35,36,37,38].

### 2.5. Outcome Variable

The outcome variable of this study was anemia among children aged 6 to 59 months. This is generated based on the cut-off hemoglobin threshold. The World Health Organization (WHO) defines anemia in children 6 to 59 months of age as hemoglobin (Hb) levels of <11.0 g/L [39,40]. The classification of anemia severity was based on the following cut-off values: not anemic (Hb ≥ 11.0 g/L), mild anemia (10.0 ≤ Hb ≤ 10.9 g/L), moderate anemia (7.0 ≤ Hb ≤ 9.9 g/L), and severe anemia (Hb < 7.0 g/L) [21,28,29,39,40].

### 2.6. Exposure Variables

Exposure variables included children’s characteristics (age, gender, birth order, received vitamin A, and child illness). For all surveys, the following variables were available: *had diarrhea recently/past 2 weeks, *had fever recently/past 2 weeks, *had cough recently/past 2 weeks. Thus, this was the basis for generating the variable “child illness”. Children’s feeding characteristics were also considered (intake of different food groups, such as *cereals, roots and tubers, *legumes and nuts, *dairy products, *flesh foods and eggs, *fruits and vegetables, and *oils and fats). To estimate the minimum dietary diversity (MDD) score, a scale based on food group intake from 0 to 6 was generated: (0) any child who did not consume any of the food groups; (1) any child who consumed one food group; (2) any child who consumed two food groups; (3) any child who consumed three food groups; (4+) any child who consumed four or more food groups. Children who consumed four or more food groups were considered to have an adequate MDD, while those who consumed fewer than four groups were considered to have an inadequate MDD.

Other characteristics included were the mother’s level of education, age of the mother at childbirth, and frequency of antenatal care visits. This study also considered household (wealth index, source of drinking water, type of toilet facility, pollution within the household, sex of household head, age of household head, whether children under 5 slept under mosquito bed net) and community factors (residence area and province). The basis for generating the variable “pollution within the household” was the variables “type of cooking fuel” and “smoking cigarette”. More details on variables can be found in the Appendix A.

### 2.7. Statistical Analysis

Data analysis was performed considering the requirements for the complex survey data. A weighting variable was generated using the sample weight variable in the DHS [41,42]. We designed a complex survey plan that considered the following variables: individual sample weight, sample strata for sampling errors/design, and cluster number [6,35,41,42,43].

### 2.8. Multilevel Mixed-Effects Analysis

We performed a bivariate analysis for all exposure variables that might be able to influence children’s anemia. Crude odds ratios were presented, and the significance threshold was fixed at a *p*-value < 0.05 (Appendix A). Multivariate models were performed for all exposure variables with a *p*-value < 0.20 in the bivariate analysis [35,42,44]. The first model (empty model) was fitted without independent variables to show the childhood anemia variance among different clusters (communities). The second model retained variables from Model 1 and incorporated maternal and child-level characteristics, including child-feeding factors. The third model retained variables from Model 1 and incorporated only household- and community-level factors. The fourth model retained variables from Model 1 and incorporated maternal, child, household, and community-level factors. Model 5 included all levels of variables (maternal, child, household, and community) along with interaction terms for child illness by age and child illness by wealth index. Survey year dummies were included in Models 2, 3, 4, and 5. For all models, the threshold of significance was fixed at a *p*-value  <  0.05 [45,46]. The adjusted odds ratio and 95% confidence intervals (CIs) are presented with the results. The random effect of the models’ standard errors (and their 95% CIs) and the inter-cluster correlation coefficient (ICC) for each model are also presented. The comparison of the models was performed by applying the Akaike information criterion (AIC). All analyses were performed using STATA version 18.0.120 software (StataCorp, College Station, TX, USA) [47].

## 3. Results

### 3.1. Study Population Characteristics

The main characteristics of study participants are described in Appendix A. The percentage of children who received vitamin A decreased from 74.2% in 2011 to 50.2% in 2022. There were changes in feeding habits, showing a decline in the percentage of children with an adequate minimum dietary diversity score (consumed four or more food groups) from 17.8% in 2011 to 4.0% in 2022. Over a decade, the prevalence of anemia in children aged 6–59 months remained high, increasing slightly from 69.1% in 2011 to 72.9% in 2022. Similarly, the prevalence of moderate, mild, and severe anemia increased slightly, from 39.1%, 26.0%, and 4.0% in 2011 to 40.6%, 28.0%, and 4.3% in 2022, respectively (Figure 1).

Table 1 shows the proportion of anemia by selected variables. A relevant proportion of children with anemia lived in rural areas, increasing from 66.1% in 2011 to 75.6% in 2022. The provinces of Nampula and Zambezia showed an increase in anemia prevalence, from 13.2% and 15.0% in 2011 to 31.7% and 17.9% in 2022, respectively. Similarly, anemia prevalence increased among children who did not receive vitamin A (from 27.7% in 2011 to 52.3% in 2022, and among children with an inadequate minimum dietary diversity score, from 23.6% in 2011 to 48.4% in 2022.

### 3.2. Determinants of Anemia Among Children

Table 2 shows the association between each factor and anemia in children aged 6–59 months. According to Model 5—which includes maternal, child, household, and community variables and has the best model fit performance (AIC of 8802)—the child’s age, having an illness, and having received vitamin A supplements play a pivotal role in childhood anemia. Furthermore, the household head’s gender, the household wealth index, and the source of drinking water significantly impact anemia prevalence (*p* < 0.05). For instance, children who received vitamin A supplements were less likely to have anemia than those who did not (aOR = 0.80, 95% CI = 0.69–0.93). Children living in female-headed households are more susceptible to anemia than those who live in male-headed households (aOR = 1.16, 95% CI = 1.01–1.32). On the other hand, children living in households that use unimproved drinking water sources are more likely to have anemia than those who live in households that use improved sources (AOR = 1.40, 95% CI = 1.19–1.65). Table 2 also presents the geographical variation in anemia among children aged 6–59 months. There is a notable clustering of anemia at both regional (ICC of 5.1%, in Model 1) and community levels within regions (ICC of 16.4%, in Model 1), indicating variability in anemia prevalence across different geographic levels.

Figure 2 presents the probability of child illness by age, while Figure 3 illustrates the probability of anemia by wealth index. Childhood anemia prevalence decreases as the children’s age increases. Specifically, children aged 6–11 months are less likely to have anemia than those aged 12–23 months (aOR = 0.75, 95% CI = 0.62–0.89) or older than 24 months (aOR = 0.36, 95% CI = 0.28–0.47).

Children who have suffered from illnesses are more likely to have anemia than those who have not (aOR = 1.44, 95% CI = 1.18–1.74). Moreover, children from richest/highest-wealth-quintile households are less likely to suffer anemia than those from poor/lower wealth quintile households (aOR = 0.69, 95% CI = 0.46–1.03).

## 4. Discussion

This study examined the determinants of anemia among children aged 6–59 months in Mozambique. Over a decade, the prevalence of anemia in Mozambican children aged 6–59 months remained high, at 69.1% in 2011 and 72.9% in 2022.

The rates reported in our study are above the average of sub-Saharan Africa (64.1%) and the prevalence of the East Africa region (56.0%), being much higher than in countries such as Malawi (63.0%), Burundi (60.9%), Rwanda (53.0%), and Ethiopia (49.3%) [9,42,48,49]. These rates are also higher than in countries such as Angola (65%), Chad (59.6%), Cape Verde (51.8%), and Egypt (52.0%) [50,51,52,53]. On the other hand, these rates are lower than those reported in some countries from West Africa such as Mali (76–88%), Guinea Bissau (80.2%), and Ghana (78.0%) [54,55,56]. Such disparities in anemia prevalence rates among these countries may be explained by geographical, climate, socioeconomic, and cultural factors [57]. Mozambique is ranked 11th out of 191 countries for being at extreme risk of climate-related hazards—due to its long coastline and geographic location [58]. Between 2006 and 2021, there hae been an increase in climate-related hazards, including cyclones, floods, and droughts—resulting in death, internal displacement, massive agricultural losses, extreme food insecurity, disease outbreaks, the destruction of assets and livelihoods, and the destruction of schools and health facilities [58]. Moreover, in recent years, consecutive climate shocks, such as droughts in the southern region of Mozambique; the Desmond, Idai, Chido, and Kenneth cyclones; and Filipo’s tropical storm, which mostly affected the central and northern regions of Mozambique, have led to an increase in acute food insecurity [59,60,61].

These recurrent natural disasters exacerbate communicable disease outbreaks (e.g., malaria, diarrhea, and cholera) and increase the cases of malnutrition among pregnant women and children [58,61]. Additionally, due to tropical storms and cyclone-related impacts, health workers cannot work properly as health facilities face shortages of medicine and supplies [61]. In such situations, areas become physically inaccessible, limiting the movement of humanitarian partners and delivery of assistance [61]. The lives of children in these regions are impacted by climate shocks, leading to deprivation, displacement, and loss of household assets, thereby increasing the burden of infectious diseases, parasitic infections, malnutrition, and anemia [58,59,60,61,62].

Our results indicate that Mozambique has shown no progress; it is even worsening, and is still far from achieving the global nutrition target by 2030 of reducing anemia rates to less than 15% in children, this remaining a severe public health problem [34,40,63,64,65]. Our findings show a slight increase in all anemia types between 2011 and 2022, with moderate and severe anemia increasing from 39.1% and 4.0% in 2011 to 40.6% and 4.3% in 2022, respectively. These rates are similar to those reported in previous studies in Mozambique that emphasize the need for more monitoring by health professionals for the early detection of anemia, particularly severe anemia, for proper referral and treatment [21]. The study findings also underscore the relevance of community-level factors in influencing health outcomes. Such persistent high rates of anemia might have an impact on Mozambique’s overall development, as they can harm school performance, productivity in adult life, and overall quality of life in general and may also lead to negative financial impacts for individuals, families, and communities [63,66].

Our study showed that the child’s age, presence of illness, having received vitamin A supplements, household head’s gender, household’s wealth index, and the source of drinking water were determinants for anemia in Mozambican children.

### 4.1. Child’s Age and Childhood Anemia

In line with our findings, previous studies [34,53,67,68,69,70,71] have also shown that anemia risk lowers as children grow and get older. This might be due to maternal anemia and poor young children’s feeding practices during the initiation of complementary feeding [72]. Some studies [73,74,75] showed that maternal iron deficiency plays a substantial role in the iron status of their infants, with mothers’ hemoglobin levels contributing to the variation in their infants’ hemoglobin concentration in the first six months of life. Children born to anemic mothers often have limited iron reserves, even when born at full term and at a typical weight [73]. On the other hand, younger children have higher iron demands, linked to early childhood development, and the lack of essential nutrients in their diet may lead to nutritional deficiencies, including iron anemia deficiency. Similarly, growth spurts coupled with the termination of lactation around 12 months of age also pose a high nutrient demand which often are not reached. As such, a good complementary diet rich in external sources of iron is mandatory, particularly during the first 1000 days of life (from pregnancy and throughout the child’s second year of life), to achieve the high iron demands in childhood [67,72,76,77]. The reinforcement of regular monitoring of anemia risk at child healthcare visits, particularly in those aged 12–23 months, could help to enable early detection and prompt intervention, contributing to reducing anemia’s prevalence [32]. Mozambique still faces huge challenges linked to the necessity of commodities in health facilities. Previous studies show that only 12.4% of health facilities, at the national level, are ready to deliver nutrition interventions related to anemia testing, and iron and folic acid supplementation through the prevention of mother-to-child transmission services [78]. Thus, more investment and coordinated support from key stakeholders can contribute greatly to overcoming this constraint and tackling children’s anemia, improving their health and development [79]. It is crucial to bolster the primary healthcare network in-country, by ensuring enough skilled staff, essential medicines, and supplies—especially for vulnerable groups like children. Strengthening this network can increase equitable coverage and access to essential health programs such as childcare and nutrition, and contribute to tackling anemia [80,81].

### 4.2. Presence of Illness and Childhood Anemia

This study’s findings concur well with previous studies that showed that children with a history of illnesses (fever, cough, or diarrhea) have a higher risk of being anemic [50,65]. This might be linked to the fact that fever is a symptom of acute febrile illness (such as malaria), resulting in anemia due to the destruction of red blood cells [50,72,82]. Additionally, infectious diseases can decrease the intake and absorption of nutrients, cause intestinal mucosa injury, and induce autoimmune reactions leading to anemia. Children who are sick often lose their appetite, which might also lead to nutritional deficiencies [82,83]. A joint statement by the WHO and the United Nations Children’s Fund (UNICEF) was issued in 2006, advising that, in settings where malaria and other infectious disease prevalence is high, iron and folic acid supplementation should be targeted at young children who are anemic and at risk of iron deficiency, in the presence of effective infectious disease control [84]. However, as mentioned before, Mozambique needs coordinated action, with the support of partners, to overcome several constraints linked with the limited readiness for delivering iron and acid folic supplementation interventions at health facilities.

### 4.3. Vitamin A Supplementation and Childhood Anemia

Similarly to earlier findings [53], we found that children who received vitamin A supplements were less likely to suffer from anemia than children who did not. The existence of a relationship between vitamin A and anemia is well-reported [85,86]. It is known that vitamin A supplementation contributes to reducing anemia’s prevalence through several mechanisms, such as the increase in the growth and differentiation of erythrocyte progenitor cells and modulation of bioavailability and the mobilization of tissue iron stores [85,86]. Vitamin A supplementation has beneficial effects for preventing the risk of death in children aged 6 to 59 months who are at risk of vitamin A deficiency, and plays a crucial role in reducing illness (e.g., new occurrences of diarrhea and measles) [87]. As such, vitamin A supplementation indirectly contributes to preventing and reducing important anemia risk factors for children (e.g., diarrhea and fever) [87,88]. Since 2008, the Ministry of Health (MoH) of Mozambique, through the technical and financial support of UNICEF and other partners, started the implementation of “National Health Weeks”, largely as a response to the low coverage of vitamin A supplementation in routine health services in previous years. Although it achieved high vitamin A supplementation coverage, the initiative is still not quite sustainable due to the high financial cost. Since 2017, the MoH has been investing efforts to strengthen routine health services, as the main vehicle for administering vitamin A capsules to all eligible children [89]. In addition to this, the MoH has been conducting efforts by creating mobile brigades with health providers or community health agents (Agentes Polivalentes de Saúde, APSs, in Portuguese) to reach children in the community, where they give lectures with messages encouraging mothers and caregivers to feed their children and babies with foods rich in vitamin A [89]. Nonetheless, the mobile brigade strategy is not always sustainable because most communities are far from health centers, combined with the problem of a lack of transport and fuel and cost allowances for health providers [89]. However, while these mobile brigades are being carried out, they can still be used to reinforce lectures on proper feeding habits focusing on preventing anemia in children.

### 4.4. Socioeconomic Status and Childhood Anemia

As suggested by previous authors [34,72,82,90,91], the evidence we found points to the fact that the wealth index plays a pivotal role in anemia, with children from the lowest-wealth-quintile households having more risk of suffering from anemia than their peers. This could be because caregivers of children from high socioeconomic classes can have better access to and afford good prevention and care health services, as they do not face major financial constraints [91]. Families from high socioeconomic classes can purchase and provide good nutritious foods, and ensure adequate dietary patterns for their children, decreasing anemia risk [91]. On the other hand, children from low-wealth quintiles often are exposed to community poverty, living in contexts with limited access to healthcare, a lack of social services, a lack of proper sanitation, and fewer job opportunities as they grow older [82]. Furthermore, our analysis period coincides with the COVID-19 pandemic, which led Mozambique to experience its first severe economic contraction in almost 30 years, affecting enterprises and households, causing income losses, and worsening living conditions [92]. In addition, other studies show that there is a high degree of poverty immobility, particularly in the northern and central regions of Mozambique, and education is a crucial factor for moving from the poor to the vulnerable group and from the vulnerable to the nonvulnerable group (across wealth quintiles) [93]. Many years of progress in the fight against poverty in Mozambique suffered a step backwards and necessitates a multisectoral approach by the government of Mozambique and its partners to revert to a downward trend for poverty rates, aiming to reduce socioeconomic inequalities and thus contribute to tackling children’s anemia [92,94]. Strategies and policies to achieve this goal include more investments in rural development, increasing the productivity of smallholder farmers, expanding markets to enable subsistence farmers to shift to commercial and sustainable agriculture, and more efforts to improve schooling and health literacy, particularly among women due to their role as children’s caregivers [92,93].

### 4.5. Gender of Household Head and Childhood Anemia

Our findings also show that the gender of the household’s head can influence childhood anemia. Children living in female-headed households are more likely to have anemia than those who live in male-headed households. This corroborates previous studies [71] which explain that mothers from poorer households often also suffer from anemia, thus increasing the risk of offspring being anemic. In such female-headed households, women have an increased number of tasks that include labor and home responsibilities. They often bear a double burden of ensuring economic security, food security, and childcare, by themselves and with no additional support [67]. The 2018 Mozambican Gender Policy and Strategy for its Implementation addresses several aspects of improving women’s access to employment in the public and private sectors [95]. However, there is an urgent need to develop specific gender-sensitive work policies and actions aimed at improving the work–life balance of women with children in care, particularly those who are household heads of children under five. In addition, more investments in care policies and social services could help women better balance childcare and work responsibilities [96], therefore, contributing to improving their health and their offspring’s well-being.

### 4.6. Water and Sanitation and Childhood Anemia

Our results corroborate previous results [50,97], showing that children living in households that use unimproved drinking water sources are more likely to have anemia than their peers, as they are more exposed to acute and chronic infections linked to increased exposure to pathogens due to environmental contamination. Several studies have demonstrated that the risk of exposure to soil-transmitted helminths (STHs) is high in settings with unimproved drinking water sources, contributing to anemia and undernutrition [98,99,100]. In such contexts, there is an increased risk of anemia and micronutrient deficiencies—resulting from inadequate intake or impaired nutrient absorption due to parasitic infections, particularly *Ascaris lumbricoides*, *Trichuris trichiura*, hookworms, and mixed infections [98,99,100]. Anemia is a serious sequela of hookworm infection due to the intestinal attachment of adult worms, this being extremely problematic in young children [98]. Policies focused on improving water sanitation, regular parasite surveillance, and integrating deworming programs with nutrition-specific interventions, particularly in low-income settings, contribute to improving children’s health and preventing anemia [97,98,99,101]. Some authors [90] argue that improved water sanitation can help to prevent infections and reduce enteropathy, leading to better iron absorption and reduced loss of nutrients through lower diarrhea prevalence, thus tackling anemia. Shreds of evidence from studies performed in Burkina Faso [102] and India [90] showed an association between access to improved water sanitation and latrines/boreholes and an improved nutritional status in children younger than 5 years, as well as better protection from enteric exposure. Kothari and collaborators [90] suggest that implementing water, sanitation, and hygiene interventions side-by-side with nutrition-specific interventions (in addition to malaria) is highly relevant for achieving success in preventing and managing anemia in high-infection-burden settings. Nonetheless, further research is required in Mozambique to explore the interaction between water sanitation (including water of household premises and water quality), parasitic infections, and childhood anemia.

## 5. Conclusions

This study revealed that childhood anemia remains a severe public health problem in Mozambique. The findings underscore the importance of community-level factors in influencing health outcomes, highlighting the urgent need for interventions that address not only individual risk factors but also the wider regional disparities that contribute to this issue.

### Recommendations and Future Approach

Tackling childhood anemia will require a set of multisectoral approaches involving the government and stakeholders, with a view to enhancing the provision of essential anemia supplies (such as vitamin A, iron, and acid folic supplements) for primary healthcare at the national level, and to reinforcing the regular monitoring of anemia risk at child healthcare visits by health providers and nutritionists, particularly in those aged 12–23 months.

Efforts should also be aimed at reducing socioeconomic inequalities through more investments in rural development and sustainable agriculture, improving water sanitation, and improving schooling and health literacy among children’s caregivers.

Re-shaping the current gender strategies for the inclusion of specific gender-sensitive work actions alongside investments in social care policies can help to improve the work–life balance of women with children in care, specifically those who are household heads of children under five.

Continuous and further research on childhood anemia (including feeding habits) in this setting can contribute hugely to leaving no one behind and reaching global nutrition goals. Future approaches in research include ensuring that data on relevant characteristics (e.g., related to feeding habits, iron supplementation, hookworm infestation, and food security) are systematically collected in future national surveys. This will contribute to the generation of stronger evidence on childhood anemia and its determinants.

## 6. Strengths and Limitations

There are several strengths linked to our study. Although there are studies on anemia in children in Mozambique, to our knowledge, this is the first study that, in addition to social, maternal, and community factors, also includes the analysis of dietary factors and anemia in children aged 6–59 months. Our study was based on two nationally representative surveys (2022–2023 DHS and 2011 DHS). The data were weighted, and genuine and reliable estimations and standard errors were provided through multivariate logistic regression. The results obtained in this study corroborate the current literature, and can greatly contribute to improving multisectoral strategies, designing interventions tailored to the needs of specific groups, and implementing a more holistic approach to reducing anemia in Mozambican children.

Some limitations of this study are linked to the fact that we used secondary data and to issues related to data collection for some variables (e.g., illness and feeding practices), which may be affected by the memory of participants, leading to recall bias and misclassification. For common morbidities (such as malaria and HIV) and other possibly relevant factors (such as worm infestation and food insecurity), direct measures were not available in the datasets. Thus, and similarly to previous studies [34,103] we used proxies for suspected malaria (e.g., children under 5 that slept under a bed net, and have had a fever recently), for worm infestation (e.g., household sanitation), and food insecurity (e.g., household wealth index).

Since child feeding practices were exclusively measured in the 2011 and 2022–2023 surveys and were missing in other similar surveys (e.g., the Immunization, Malaria and HIV/AIDS Indicator Survey 2015 and the Malaria Indicator Survey 2018), the pooled data analysis was based only on the 2011 and 2022–2023 DHS sample data [28,29]. The previous history of iron supplementation in children aged 6–59 months was not taken into consideration by our analysis, as such information was only reported in the 2011 survey and was missing in the other surveys (2022–2023 DHS, 2018 AIS, and 2015 MIS). Ensuring the systematic collection, throughout the years of surveys, of this and other missing information (e.g., child feeding practices, iron supplementation during pregnancy) can be of great relevance for further research on children’s anemia in this setting.

This study did not include a detailed cost–benefit analysis or prioritization of interventions that could have enriched the decision-making process. Nonetheless, this falls outside of the scope of this study. In addition, the design of the surveys used (2022–2023 and 2011 DHS) did not provide enough data to perform accurate cost–benefit analyses or comparisons between interventions (e.g., costs of vitamin A or iron supplementation interventions, child anemia-related mortality, or reduction in disability-adjusted life years). Previous studies [104] conducted by the Government of Mozambique and key partners have assessed the cost-effectiveness of nutrition interventions (including vitamin A and iron supplementation), applying additional data sources and specialized methodologies, including qualitative research. Further research could explore this issue to provide stronger and updated evidence on specific anemia interventions, which could be useful for programmatic decision-making.

## Figures and Tables

**Figure 1 healthcare-13-00635-f001:**
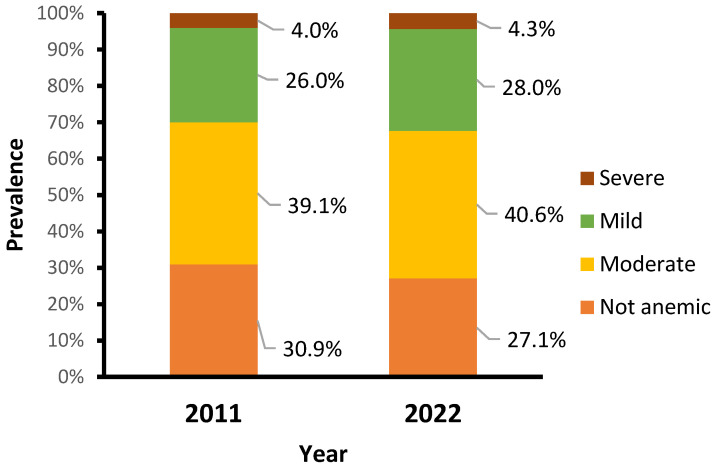
Prevalence of anemia by year.

**Figure 2 healthcare-13-00635-f002:**
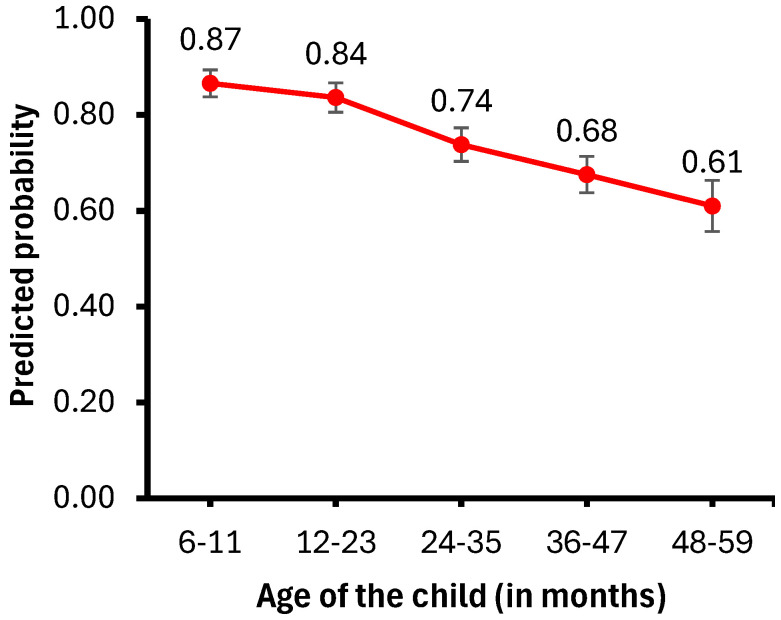
Predicted probability of child illness by child’s age.

**Figure 3 healthcare-13-00635-f003:**
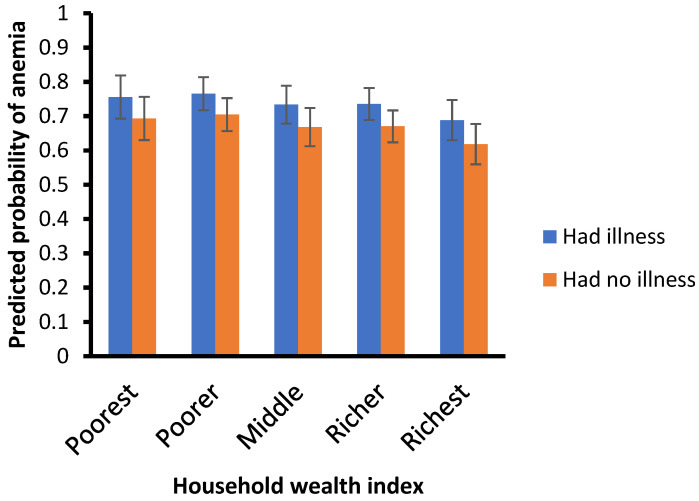
Predicted probabilities of anemia by wealth index.

**Table 1 healthcare-13-00635-t001:** Proportion of anemia by selected categorical variables (DHS 2011 DHS2022) (N = 8143).

Variable	2011 (n = 4597)					2022 (n = 3546)				
	Overall	Severe	Moderate	Mild	Not Anemic	Overall	Severe	Moderate	Mild	Not Anemic
Child’s age (in months)										
6–11	15.1	25.1	15.2	13.6	8.0	13.4	16.5	14.2	11.8	4.8
12–23	28.1	38.1	30.6	22.8	17.4	26.3	38.7	27.0	23.3	13.9
24–35	20.7	17.9	20.4	21.6	22.7	24.4	28.3	24.6	23.6	22.2
36–47	19.7	16.1	19.5	20.5	25.3	18.9	9.7	18.5	20.9	26.7
48–59	16.3	2.8	14.3	21.6	26.6	17.0	6.8	15.8	20.4	32.5
Gender of child										
Male	49.4	47.1	51.8	46.0	49.8	48.3	51.9	49.4	46.2	47.2
Female	50.6	52.9	48.2	54.0	50.2	51.7	48.1	50.6	53.8	52.8
Child illness										
No	74.6	64.2	73.7	77.5	82.4	81.3	78.0	81.1	82.0	84.7
Yes	25.4	35.8	26.3	22.5	17.6	18.7	22.0	18.9	18.0	15.3
Children aged 6–59 months given vit. A supplement										
No	27.7	28.9	30.2	24.0	21.3	52.3	66.1	54.4	47.1	43.2
Yes	72.3	71.1	69.8	76.0	78.7	47.7	33.9	45.6	52.9	56.8
Feeding characteristics										
Feeding: Group 1 (cereals, roots, and tubers)										
No	35.9	31.0	35.2	37.7	40.8	54.0	55.1	52.4	56.2	64.2
Yes	64.1	69.0	64.8	62.3	59.2	46.0	44.9	47.6	43.8	35.8
Feeding: Group 2 (legumes and nuts)										
No	75.8	77.7	75.1	76.5	81.0	87.9	85.7	88.1	87.9	87.8
Yes	24.2	22.3	24.9	23.5	19.0	12.1	14.3	11.9	12.1	12.2
Feeding: Group 3 ((dairy products (milk, yogurt, cheese))										
No	90.5	91.1	91.1	89.4	89.5	95.4	94.6	96.2	94.2	95.4
Yes	9.5	8.9	8.9	10.6	10.5	4.6	5.4	3.8	5.8	4.6
Feeding: Group 4 [flesh foods (meat, fish, fowl, liver, other organs, and eggs)]										
No	61.9	59.8	61.9	62.3	66.1	80.3	82.5	80.2	80.0	85.4
Yes	38.1	40.2	38.1	37.7	33.9	19.7	17.5	19.8	20.0	14.6
Feeding: Group 5 (fruits and vegetables)										
No	46.2	39.3	45.8	47.9	50.8	74.4	75.8	73.6	75.4	80.7
Yes	53.8	60.7	54.2	52.1	49.2	25.6	24.2	26.4	24.6	19.3
Feeding: Group 6 (oils and fats)										
No	73.3	77.9	74.0	71.5	73.1	-	-	-	-	-
Yes	26.7	22.1	26.0	28.5	26.9	-	-	-	-	-
Feeding diversity score										
0	29.7	23.6	28.5	32.4	37.0	50.1	48.4	48.4	52.7	62.5
1	9.1	13.2	9.7	7.6	7.4	11.7	14.5	12.3	10.6	6.9
2	26.4	27.7	27.3	24.9	23.6	22.4	23.6	24.5	19.4	16.7
3	16.3	14.2	16.2	16.8	15.9	11.8	10.0	11.2	12.9	9.8
4+	18.5	21.4	18.4	18.3	16	3.9	3.5	3.6	4.5	4.1
Caregivers’ characteristics										
Education level										
No education	38.5	41.5	39.4	36.8	33.1	33.7	39.3	34.3	31.9	28.2
Primary	52.7	51.3	53.8	51.3	49.7	49.0	58.4	51.5	44.0	45.5
Secondary/Higher	8.8	7.3	6.8	11.9	17.2	17.3	2.2	14.2	24.1	26.3
At least 4 ANC visits										
Fewer than 4 visits	81.7	82.0	83.1	79.5	79.8	84.2	91.7	84.5	82.7	89.2
4+ visits	18.3	18.0	16.9	20.5	20.2	15.8	8.3	15.5	17.3	10.8
Household characteristics										
Wealth index										
Poorest	27.0	44.4	29.9	20.0	17.0	29.2	48.8	32.4	21.6	20.1
Poorer	25.1	29.1	25.7	23.4	17.6	24.5	31.5	26.7	20.3	18.9
Middle	18.4	14.6	18.6	18.8	20.1	19.2	12.7	17.5	22.8	21.0
Richer	18.7	9.6	16.7	23.2	23.2	17.4	5.5	15.2	22.4	20.9
Richest	10.7	2.2	9.1	14.6	22.2	9.7	1.7	8.2	13.1	19.0
Sex of household head										
Male	71.0	70.8	71.8	70.0	71.4	75.8	81.9	75.6	75.1	75.2
Female	29.0	29.2	28.2	30.0	28.6	24.2	18.1	24.4	24.9	24.8
Source of drinking water										
Improved	30.3	17.2	26.4	38.2	46.4	52.3	38.3	49.0	59.4	65.6
Unimproved	69.7	82.8	73.6	61.8	53.6	47.7	61.7	51.0	40.6	34.4
Type of toilet facility										
Improved	20.2	12.0	18.0	24.6	29.7	24.6	19.7	22.6	28.2	28.6
Unimproved	79.8	88.0	82.0	75.4	70.3	75.4	80.3	77.4	71.8	71.4
All children under 5 slept under a mosquito net										
No	64.5	65.7	65.1	63.3	64.5	60.2	55.0	60.8	60.0	62.8
Yes	35.5	34.3	34.9	36.7	35.5	39.8	45.0	39.2	40.0	37.2
Pollution within household										
No	1.3	0.3	0.8	2.1	2.7	1.5	0.0	1.1	2.4	5.1
Yes	98.7	99.7	99.2	97.9	97.3	98.5	100.0	98.9	97.6	94.9
Community factors										
Residence area										
Urban	33.9	10.7	21.1	27.5	33.9	24.4	6.2	21.6	31.3	36.2
Rural	66.1	89.3	78.9	72.5	66.1	75.6	93.8	78.4	68.7	63.8
Province										
Niassa	7.0	6.6	5.7	5.0	7.0	7.0	2.9	6.0	9.2	13.3
Cabo Delgado	6.5	11.7	10.2	7.1	6.5	7.2	7.7	8.0	6.0	4.9
Nampula	13.2	17.7	16.4	16.0	13.2	31.7	55.8	35.5	22.5	16.1
Zambezia	15.0	42.5	26.9	19.8	15.0	17.9	20.0	18.6	16.6	15.5
Tete	13.4	4.9	12.2	14.2	13.4	9.7	3.7	8.7	12.0	13.7
Manica	8.3	7.3	7.3	7.8	8.3	7.3	0.7	5.3	11.2	9.2
Sofala	11.6	3.1	8.0	10.6	11.6	7.6	8.2	8.4	6.4	5.8
Inhambane	5.7	1.7	3.9	5.5	5.7	3.1	0.7	2.9	3.6	4.3
Gaza	6.0	2.9	4.3	4.5	6.0	3.4	0.4	3.0	4.6	4.2
Maputo provincia	8.1	0.7	2.9	6.1	8.1	3.9	0.0	3.0	5.9	8.6
Maputo cidade	5.2	1.0	2.2	3.4	5.2	1.1	0.0	0.6	2.1	4.4

**Table 2 healthcare-13-00635-t002:** Determinants of anemia among children aged 6–59 months.

Variable	Model 1 (Empty)	Model 2 aOR [95%CI]	Model 3 aOR [95%CI]	Model 4 aOR [95%CI]	Model 5 aOR [95%CI]
Age of the mother		0.99 [0.98, 1.01]		0.99 [0.98, 1.01]	0.99 [0.98, 1.01]
Education level					
No education ^†^		1.00		1.00	1.00
Primary		0.95 [0.81, 1.11]		1.02 [0.86, 1.20]	1.02 [0.86, 1.21]
Secondary/Higher		0.60 [0.48, 0.76] **		0.78 [0.56, 1.08]	0.77 [0.55, 1.08]
At least 4 ANC visits					
Fewer than 4 visits ^†^		1.00		1.00	1.00
4+ visits		0.98 [0.78, 1.22]		1.01 [0.80, 1.28]	1.01 [0.80, 1.28]
Child’s age (in months)					
6–11 ^†^		1.00		1.00	1.00
12–23		0.77 [0.61, 0.96] *		0.77 [0.62, 0.96] *	0.75 [0.62, 0.89] **
24–35		0.40 [0.31, 0.51] **		0.40 [0.31, 0.51] **	0.36 [0.28, 0.47] **
36–47		0.28 [0.23, 0.34] **		0.28 [0.23, 0.35] **	0.26 [0.20, 0.32] **
48–59		0.20 [0.16, 0.25] **		0.20 [0.16, 0.26] **	0.20 [0.17, 0.25] **
Child illness					
No ^†^		1.00		1.00	1.00
Yes		1.40 [1.14, 1.72] **		1.44 [1.18, 1.75] **	1.44 [1.18, 1.74] **
Children aged 6–59 months given vit. A supplement					
No ^†^		1.00		1.00	1.00
Yes		0.77 [0.65, 0.92] **		0.80 [0.69, 0.94] **	0.80 [0.69, 0.93] **
Feeding diversity score				0.99 [0.95, 1.02]	
0		1.00		1.00	1.00
1		1.08 [0.77, 1.51]		1.04 [0.73, 1.48]	1.04 [0.72, 1.50]
2		0.90 [0.73, 1.11]		0.88 [0.71, 1.10]	0.87 [0.69, 1.09]
3		0.92 [0.76, 1.12]		0.92 [0.75, 1.13]	0.91 [0.74, 1.12]
4+		0.98 [0.86, 1.12]		0.99 [0.85, 1.16]	0.98 [0.84, 1.14]
Wealth index					
Poorest ^†^			1.00	1.00	1.00
Poorer			1.04 [0.90, 1.20]	1.06 [0.90, 1.26]	1.02 [0.84, 1.23]
Middle			0.82 [0.67, 1.02]	0.87 [0.71, 1.07]	0.83 [0.69, 1.01]
Richer			0.81 [0.67, 0.98] *	0.88 [0.74, 1.06]	0.84 [0.71, 1.00]
Richest			0.59 [0.43, 0.81] **	0.67 [0.47, 0.97] *	0.69 [0.46, 1.03]
Sex of household head					
Male ^†^			1.00	1.00	1.00
Female			1.12 [0.98, 1.28]	1.15 [1.01, 1.31] *	1.16 [1.01, 1.32] *
Source of drinking water					
Improved ^†^			1.00	1.00	1.00
Unimproved			1.38 [1.19, 1.59] **	1.41 [1.19, 1.67] **	1.40 [1.19, 1.65] **
Type of toilet facility					
Improved ^†^			1.00	1.00	1.00
Unimproved			0.98 [0.72, 1.33]	0.97 [0.71, 1.32]	0.96 [0.71, 1.31]
Residence area					
Urban ^†^			1.00	1.00	1.00
Rural			1.12 [0.95, 1.33]	1.11 [0.92, 1.33]	1.11 [0.93, 1.34]
Child’s age # child illness					
6–11 # No illness					1.00
6–11 # Had illness					1.00 [0.99, 1.00]
12–23 # No illness					1.00
12–23 # Had illness					1.20 [0.76, 1.90]
24–35 # No illness					1.00
24–35 # Had illness					1.52 [1.01, 2.29] *
36–47 # No illness					1.00
36–47 # Had illness					1.61 [1.03, 2.52] *
48–59 # No illness					1.00
48–59 # Had illness					0.85 [0.53, 1.37]
Wealth index # child illness					
Poorest # No illness					1.00
Poorest # Had illness					0.99 [0.99, 1.00]
Poorer # No illness					1.00
Poorer # Had illness					1.29 [0.89, 1.88]
Middle # No illness					1.00
Middle # Had illness					1.25 [1.08, 1.45] **
Richer # No illness					1.00
Richer # Had illness					1.32 [0.94, 1.86]
Richest # No illness					1.00
Richest # Had illness					0.93 [0.54, 1.61]
Survey year					
2011		1.00	1.00	1.00	1.00
2022		1.17 [0.95, 1.45]	1.23 [0.97, 1.55]	1.28 [1.06, 1.56] *	1.28 [1.05, 1.55] *
Random effect					
Regional-level variance (SE)	0.20 (0.08)	0.16(0.05)	0.08 (0.03)	0.09(0.03)	0.09(0.03)
Region > community level variance (SE)	0.44 (0.15)	0.49(0.17)	0.39(0.13)	0.47(0.16)	0.47(0.16)
Regional ICC (%)	5.1	4.0	2.2	2.4	2.4
Community|region ICC (%)	16.4	16.5	12.6	14.6	14.6
Model fit statistics					
Log likelihood	−4.675	−4.430	−4.627	−4.399	−4.391
AIC	9.355	8.881	9.274	8.818	8.802

Notes: ^†^ is a reference category; aOR: adjusted odds ratio; SE: standard error; ICC: intra-cluster correlation coefficient; AIC: Akaike Information Criterion; 95% confidence intervals (CIs) in brackets; # represents interaction effects; * *p* < 0.05, ** *p* < 0.01.

## Data Availability

Data generated during this study are available in the article and its Appendix A.

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
