# Peer review of "Childhood Anemia in Mozambique: A Multilevel Mixed-Effects Analysis of 2011–2022/23 Population-Based Surveys"

_healthcare, 2025, doi:10.3390/healthcare13060635_

Round 1
Reviewer 1 Report
Comments and Suggestions for Authors
Based on four large-scale representative datasets, this study outlined the overall prevalence (71.4%) of anemia among children aged 6-59 months in the Republic of Mozambique, and made a comprehensive examination of various levels of determinants, such as histories of illness and receiving Vitamine A supplements, and household wealth.
Overall, the study was based on solid data analysis and the manuscript was clearly written. I only propose a few minor suggestions.
1. The time trend: Is it possible to include survey year dummies in your model, on one hand, to control time to obtain net effects of other factors; and on the other hand, to delineate the possible time trend? For example, the overall prevalence of anemia is 71.4 percent, but what was it in 2011 and then in 2022?
2. I noticed that there is some basic information for this in your supplementary file. Maybe you could consider bringing one descriptive table in the supplementary file to the main text.
3. Table 2: The first column might be necessary. Without Table 1, the first column of Table 2 would be vital. Anyway, in your present Table 2, you mentioned ICC in the note, but I cannot locate ICC info in the Table.
4. Figure 1: the x-axis should be clearly labelled with age in months.
Author Response
Dear Reviewer 1,
Thank you very much for taking the time to review this manuscript. Please find the detailed responses below:
Comment 1: Based on four large-scale representative datasets, this study outlined the overall prevalence (71.4%) of anemia among children aged 6-59 months in the Republic of Mozambique and made a comprehensive examination of various levels of determinants, such as histories of illness and receiving Vitamine A supplements, and household wealth.
Overall, the study was based on solid data analysis, and the manuscript was written. I only propose a few minor suggestions.
Response 1: Thank you for your comments and for taking the time to review this manuscript.
Comment 2: The time trend: Is it possible to include survey year dummies in your model, on the one hand, to control time to obtain net effects of other factors; and on the other hand, to delineate the possible time trend? For example, the overall prevalence of anemia is 71.4 percent, but what was it in 2011 and then in 2022?
Response 2: Thank you, we revised it. Survey year dummies for the model, as well as a time trend explanation were included. Please see lines 229-233, Figure 1, Table 2 and Supplementary file Table 3.
Comment 3: I noticed that there is some basic information for this in your supplementary file. Maybe you could consider bringing one descriptive table in the supplementary file to the main text.
Response 3: Thank you, we revised it. One descriptive table from the supplementary file was added to the main text, please see Table 1. Nonetheless, we chose to maintain the Supplementary file Table 3 outside the main manuscript (due to its size).
Comment 4: Table 2: The first column might be necessary. Without Table 1, the first column of Table 2 would be vital. Anyway, in your present Table 2, you mentioned ICC in the note, but I cannot locate ICC info in the Table.
Response 4: Thank you, both tables were revised, and ICC information was included in Table 2.
Comment 5: Figure 1: the x-axis should be clearly labeled with age in months.
Response 5: Thank you. We revised it and corrected the label. Please see Figure 2 (previously labeled as Figure 1).
Reviewer 2 Report
Comments and Suggestions for Authors
The authors studied the “Childhood anemia in Mozambique: What are the determinants? A multilevel mixed effects analysis of 2011-2022/23 population-based surveys.”. However, I have the following comments, suggested edits, and questions.
Abstract:
This study aims to evaluate using different determinants for Childhood anemia in Mozambique. It would be better if the authors mentioned the conclusion based on their findings.
Introduction:
In the last paragraph, kindly develop an argument about the importance of study leading to study objectives. Also, please mention causes of anemia and any previous study evaluated the determinants of anemia. (please improve introduction section)
Methodology:
kindly explain inclusion criteria precisely with a), b), c) description for better understanding. It is better to add a subheading of “inclusion/exclusion criteria” for the study.
Variables: The authors considered determinants of anemia in children; however, I am wondering about the previous history of iron supplementation was taken into consideration. If not, please mention it in the limitation section.
Please include 1 subheading explaining the futuristic approach and recommendations to reduce the risk of childhood anemia in Mozambique
Author Response
Dear Reviewer 2,
Thank you very much for taking the time to review this manuscript. Please find the detailed responses below:
Commment 1: Abstract:
This study aims to evaluate using different determinants for Childhood anemia in Mozambique. It would be better if the authors mentioned the conclusion based on their findings.
Response 1: Thank you, we revised it, please see lines 37-42.
Comment 2: Introduction:
In the last paragraph, kindly develop an argument about the importance of study leading to study objectives. Also, please mention the causes of anemia and any previous studies that evaluated the determinants of anemia. (please improve the introduction section)
Response 2: Thank you. We revised it, and the introduction section was improved: an argument about the importance of the study leading to its objectives was developed (please see lines 91-98). The causes of anemia (such as chronic blood loss, poor diets, micronutrient deficiencies, infectious diseases,...) were mentioned, and any previous study that evaluated the determinants of anemia in Mozambique was mentioned (please see lines 56-72, and lines 91-98).
Comment 3: Methodology:
Kindly explain inclusion criteria precisely with a), b), c) descriptions for better understanding. It is better to add a subheading of "inclusion/exclusion criteria" for the study.
Response 3: Thank you. We revised it. In the DHS 2011 and 2022-2023, child anemia measurements were performed only in children aged 6–59 months. As such, this study included information on children: a) who were aged 6–59 months (at the time of the surveys); b) had undergone anemia testing and had data available on hemoglobin (Hb) determination. Please note that we prefer entitling the subheading as "Selection of study participants" instead of "inclusion/exclusion criteria", for better reader understanding (please see lines 156-160).
Comment 4: Variables: The authors considered determinants of anemia in children; however, I am wondering if the previous history of iron supplementation was taken into consideration. If not, please mention it in the limitation section.
Response 4: Thank you. The previous history of iron supplementation wasn´t taken into consideration as this information was not well captured by the surveys, for instance, only one survey (DHS 2011) included information on iron supplementation by children aged 6-59 months (reported as children who received iron supplements previously). As recommended, such limitation is now described in the limitation section, please see lines 516-522.
Comment 5: Please include 1 subheading explaining the futuristic approach and recommendations to reduce the risk of childhood anemia in Mozambique
Response 5: Thank you. We revised it, and one subheading explaining the recommendations and futuristic approach was included, please see lines 473-493.
Reviewer 3 Report
Comments and Suggestions for Authors
Dear authors,
-As improvement could perhaps choose fewer critical variables, such as presence of malaria, parasitic infections and malnutrition, were addressed with proxies (e.g., use of mosquito nets), as this limits the precision of the results.
-Data are cross-sectional, which precludes establishing direct causal relationships between the factors analysed and anaemia.
-Differences in methodology and quality of collection between surveys (2011-2023) may introduce biases. Perhaps they could limit the sample analysis to fewer years.
Author Response
Dear Reviewer 3,
Thank you very much for taking the time to review this manuscript. Please find the detailed responses below:
Comment 1: Dear authors, As improvement could perhaps choose fewer critical variables, such as presence of malaria, parasitic infections and malnutrition, were addressed with proxies (e.g., use of mosquito nets), as this limits the precision of the results.
Answer 1: Thank you. While we acknowledge the importance of including critical variables, direct measures for common morbidities such as malaria and worm infestation were not available in the dataset. Thus, and in line with previous literature, we used proxies to mitigate this limitation. We have explicitly acknowledged this constraint in the Strengths and Limitations section of the manuscript, please see lines 525-530.
Comment 2: -Data are cross-sectional, which precludes establishing direct causal relationships between the factors analysed and anaemia.
Answer 2: Thank you. Due to the cross-sectional study design, the data analyzed allow us to assess associations but not establish causal relationships (e.g., whether anemia precedes the illnesses or vice versa). Therefore, we can determine whether children who experienced illnesses (as reported during data collection) also had anemia, but we cannot infer causation. While we acknowledge this concern, we have opted not to include it in the limitations section.
Comment 3: -Differences in methodology and quality of collection between surveys (2011-2023) may introduce biases. Perhaps they could limit the sample analysis to fewer years.
Answer 3: Thank you. We acknowledge the limitations linked to data quality and collection in these types of surveys. In the Limitations section, we explained that we used secondary data and highlighted potential issues related to data collection for some variables (e.g., illness and feeding practices), which may be affected by participant recall bias and misclassification (please see lines 523–525).
As suggested by reviewer 1, we limited our analysis to 2011 DHS and 2022/23 DHS, excluding the 2015 and 2018 surveys. We chose to retain these two surveys because our study aimed to perform a comprehensive analysis of feeding and other determinants that may influence childhood anemia. Please note that feeding practices were only measured in these two surveys (2011 and 2022/23)(please see lines 531-534).
Reviewer 4 Report
Comments and Suggestions for Authors
Thank you for your efforts, especially for the statistical analysis of this huge data
Comments
1-The title will be better if it omits the question
2-How to make sure that any of these children were not included twice from the other database and that there is duplication of the number of cases?
Also, if the child was included in different years
3- Children who have suffered from illnesses were more likely to 34 have anemia than those who have not. Why not anemia among those children cause illnesses?
4- In the methods: elaborate on the following models as you only describe the content of the model 5
Model 1 (Empty)
Model 2
Model 3
Model 4
5-Figure 1 Predictive margins of child illness interacting with age
Add age in months
Author Response
Dear Reviewer 4,
Thank you very much for taking the time to review this manuscript. Please find the detailed responses below:
Comment: Thank you for your efforts, especially for the statistical analysis of this huge data.
Response: Thank you for your comments and the time dedicated to reviewing this manuscript.
Comment 1-The title will be better if it omits the question
Response 1: Thank you, we revised the title, and it omits the question now.
Comment 2-How to make sure that any of these children were not included twice from the other database and that there is duplication of the number of cases? Also, if the child was included in different years.
Response 2: Thank you for raising this important concern. The time interval between the two surveys is approximately 11 years. Given that data collection specifically targets children aged 6–59 months, it is not possible for the same children included in the 2011 survey to appear in the 2022 survey. The non-overlapping age range ensures that no individual child could be eligible for inclusion in both survey periods, effectively eliminating the risk of duplication across the datasets.
Comment 3- Children who have suffered from illnesses were more likely to have anemia than those who have not. Why not anemia among those children cause illnesses?
Response 3: Thank you. Due to the study design (of the surveys), the available data only allows us to assess possible correlations or associations, but not causality (e.g., whether anemia occurs before the illnesses or vice versa). As such, we can determine whether children who suffered from illnesses (as reported during data collection) had anemia or not, but we cannot establish a causal relationship.
Comment 4- In the methods: elaborate on the following models as you only describe the content of Model 5 Model 1 (Empty) Model 2 Model 3 Model 4
Response 4: Thank you. We revised it, please see lines 208-217.
Comment 5-Figure 1 Predictive margin of child illness interacting with age. Add age in months
Response 5: Thank you. We revised it and corrected the label. Please see Figure 2 (previously labeled as Figure 1).
Reviewer 5 Report
Comments and Suggestions for Authors
The manuscript provides a comprehensive analysis of the determinants of childhood anemia in Mozambique and highlights critical areas for intervention. However, addressing the noted weaknesses, particularly in terms of presentation clarity, depth of discussion, and policy implications, could significantly enhance its impact. By refining these aspects, the study could make a stronger contribution to the literature and guide effective public health strategies in Mozambique.
Comments:
1. While the topic is important, the manuscript does not introduce significantly new methodologies or groundbreaking findings compared to existing literature on anemia determinants
2. Reliance on secondary data introduces potential biases, including recall bias for variables like illness and feeding practices. Moreover, the exclusion of key variables such as direct measures of food insecurity and helminth infections limits the scope of the findings
3. The results section is dense and could benefit from clearer organization, particularly when discussing statistical models. For example, some tables (e.g., Table 2) are not sufficiently integrated into the narrative, making it harder for readers to connect the text with the data.
4. While the manuscript identifies determinants, it does not provide a detailed cost-benefit analysis or prioritization of interventions. For example, the effectiveness of existing strategies like vitamin A supplementation could be more critically evaluated.
5. The discussion briefly mentions geographic and climatic factors but does not explore these in depth. For example, how do natural disasters or regional disparities specifically influence anemia prevalence?
Author Response
Dear Reviewer 5,
Thank you very much for taking the time to review this manuscript. Please find the detailed responses below:
Comment: The manuscript provides a comprehensive analysis of the determinants of childhood anemia in Mozambique and highlights critical areas for intervention. However, addressing the noted weaknesses, particularly in terms of presentation clarity, depth of discussion, and policy implications, could significantly enhance its impact. By refining these aspects, the study could make a stronger contribution to the literature and guide effective public health strategies in Mozambique.
Response: Thank you for your comments. We also appreciate your time spent reviewing the manuscript. We incorporated the recommended corrections.
Comment 1: While the topic is important, the manuscript does not introduce significantly new methodologies or groundbreaking findings compared to existing literature on anemia determinants.
Response 1: Thank you for your comments regarding the importance of the topic of this manuscript. While the methodologies and findings may not seem groundbreaking, we strongly believe this study provides relevant insights into anemia among Mozambican children. Thus, addressing the local need for evidence can help guide interventions (led by policymakers, children's health programs, and health professionals) toward anemia prevention and contribute to its reduction in this setting.
Comment 2: Reliance on secondary data introduces potential biases, including recall bias for variables like illness and feeding practices. Moreover, the exclusion of key variables such as direct measures of food insecurity and helminth infections limits the scope of the findings
Response 2: Thank you for your comments. Despite the exclusion of key variables in the surveys (such as direct measures of food security and helminth infections), we believe that the use of proxies applied in previous literature (e.g. household sanitation for worm infestation and household wealth index for food insecurity) may contribute overcoming related challenges. Overall, we agree with these comments, thus such limitations were acknowledged as study limitations, in the ´Strengths and Limitations´ section of this manuscript.
Comment 3: The results section is dense and could benefit from clearer organization, particularly when discussing statistical models. For example, some tables (e.g., Table 2) are not sufficiently integrated into the narrative, making it harder for readers to connect the text with the data.
Response 3: Thank you for the feedback. We have revised the Results section to improve its clarity and organization. Specifically, we have restructured the narrative to better integrate the tables, including Table 2, into the text. This ensures a stronger connection between the data presented and the discussion, making it easier for readers to follow the findings and their interpretation. We hope these revisions address your Please see Table 1 (and lines 238-244) and Table 2 (lines 252-267).
Comment 4: While the manuscript identifies determinants, it does not provide a detailed cost-benefit analysis or prioritization of interventions. For example, the effectiveness of existing strategies like vitamin A supplementation could be more critically evaluated.
Response 4: Thank you for your comment. While we acknowledge that a detailed cost-benefit analysis or prioritization of interventions would add value to the decision-making process, it falls behind the scope of this study. The design of the surveys used (DHS) does not provide enough information to perform accurate cost-benefit analyses or comparisons between interventions (e.g. costs of vitamin A or iron supplementation interventions, children anemia-related mortality, reduction in disability-adjusted life years). Additionally, previous studies conducted by the Government of Mozambique and key partners already explored the cost-effectiveness of nutrition interventions (including vitamin A and iron supplementation), applying additional data sources and specialized methodologies. We included this information in the limitation section.
Comment 5: The discussion briefly mentions geographic and climatic factors but does not explore these in depth. For example, how do natural disasters or regional disparities specifically influence anemia prevalence?
Response 5: Thank you for your valuable comment. We have revised the Discussion section to address this point in more depth. Additional information on how natural disasters and regional disparities influence anemia prevalence has been included. Specifically, we expanded on these aspects in lines 115-119 and lines 298-307. We hope these additions provide the clarity and depth requested.
Round 2
Reviewer 2 Report
Comments and Suggestions for Authors
Thank you for considering my suggestions and authors have now significantly improved the manuscript. However, consider few more suggestions to improve the manuscript.
- Discussion section:
- subheading Water and Sanitation and Childhood Anemia:
- Authors have well explained the relationship between different variables and anemia, however, I will suggest to add studies related to parasitic infection and anemia, as many studies has concluded that soil-transmitted helminthiasis in children can lead to micronutrient deficiencies such as (doi: 10.1111/pim.13015)
Author Response
Comment: Thank you for considering my suggestions and authors have now significantly improved the manuscript. However, consider few more suggestions to improve the manuscript.
- Discussion section:
- subheading Water and Sanitation and Childhood Anemia:
- Authors have well explained the relationship between different variables and anemia, however, I will suggest to add studies related to parasitic infection and anemia, as many studies has concluded that soil-transmitted helminthiasis in children can lead to micronutrient deficiencies such as (doi: 10.1111/pim.13015)
Response: Thank you for your time and valuable comments to improve this manuscript. We revised it accordingly, incorporating additional studies related to parasitic infections and anemia, including the suggested reference. We highlighted that several studies demonstrate a high risk of soil-transmitted helminths (STHs) in areas with unimproved drinking water sources, contributing to anemia and undernutrition. Additionally, we emphasized that in these contexts, there is an increased risk of anemia and micronutrient deficiencies, resulting from inadequate intake or impaired nutrient absorption due to parasitic infections, particularly Ascaris lumbricoides, Trichuris trichiura, hookworms, and mixed infections. Furthermore, we emphasized the need for policies focused on: improving water sanitation, regular parasite surveillance, and integrating deworming programs with nutrition-specific interventions, especially in low-income settings, to enhance children's health and prevent anemia. Please see lines 461-471, and 482.
Reviewer 5 Report
Comments and Suggestions for Authors
- The new version of the manuscript partially addresses points 1, 3, 4, and point 5 are acknowledged but not deeply resolved.
- Point 2 (secondary data limitations) is aknowledged but not mitigated .
For a more substantial version please better integrate tables into the narrative, provide a clearer discussion of regional/climatic factors, and attempt, at least, a qualitative assessment of intervention prioritization.
Author Response
Comment:
- The new version of the manuscript partially addresses points 1, 3, 4, and point 5 are acknowledged but not deeply resolved.
- Point 2 (secondary data limitations) is aknowledged but not mitigated .
For a more substantial version please better integrate tables into the narrative, provide a clearer discussion of regional/climatic factors, and attempt, at least, a qualitative assessment of intervention prioritization.
Response: Thank you for your time and valuable comments to improve this manuscript.
- Comment: Point 2 (secondary data limitations) is acknowledged but not mitigated.
Response: Thank you. While the DHS dataset may have limitations, it is globally recognized as a credible source of data for health and socio-economic indicators in low- and middle-income countries. As with any other social indicator survey (e.g., DHS, AIS, MICS), we acknowledge these limitations, but there is nothing else we can do to mitigate the limitations beyond the variables we are using in this study.
- Comment: For a more substantial version please better integrate tables into the narrative, provide a clearer discussion of regional/climatic factors,and attempt, at least, a qualitative assessment of intervention prioritization.
Response: Thank you. We revised the discussion of regional(climatic factors). We have clarified that Mozambique ranks 11th in extreme climate risk due to its coastline and geographic location. Between 2006 and 2021, climate-related hazards, including cyclones, floods, and droughts, have caused deaths, displacement, massive agricultural losses, food insecurity, and health facility destruction. Recurrent climate shocks, such as cyclones and droughts, have worsened food insecurity and exacerbated malnutrition and communicable disease outbreaks, especially among pregnant women and children. These disasters also limit health services and humanitarian assistance. As a result, climate shocks significantly impact children’s lives by causing deprivation and displacement and increasing the burden of anemia and infectious diseases. Please see lines 300-318.
Regarding the other suggestion, we are unsure whether the reviewer is requesting us to do a qualitative assessment and have the results integrated into the study. Nonetheless, we have discussed the limitations of this study and proposed future research directions (including studies with specialized methodologies, such as qualitative assessment). We understand that a qualitative assessment would require separate study protocols and local ethics approval.